# Acute kidney injury and acute kidney recovery following Transcatheter Aortic Valve Replacement

**Marilou Peillex[1], Benjamin Marchandot[1], Kensuke Matsushita[1,2], Eric Prinz[3], Sebastien Hess[1], Antje Reydel[1], Marion Kibler[1], Adrien Carmona[1], Antonin Trimaille[1], Joe Heger[1], Hélène Petit-Eisenmann[1], Annie Trinh[1], Laurence Jesel[1,2], Patrick Ohlmann[1], Olivier Morel[1,2]***

**1** Division of Cardiovascular Medicine, Nouvel Hôpital Civil, Strasbourg University Hospital, Strasbourg, France, **2** INSERM (French National Institute of Health and Medical Research), UMR 1260, Regenerative Nanomedicine, FMTS, Strasbourg, France, **3** Department of Nephrology, Nouvel Hôpital Civil, Strasbourg University Hospital, Strasbourg, France

\* olivier.morel@chru-strasbourg.fr

## Abstract

### Background

Acute kidney injury (AKI) is associated with a dismal prognosis in Transcatheter Aortic Valve replacement (TAVR). Acute kidney recovery (AKR), a phenomenon reverse to AKI has recently been associated with better outcomes.

### Methods

Between November 2012 to May 2018, we explored consecutive patients referred to our Heart Valve Center for TAVR. AKI was defined according to the VARC-2 definition. Mirroring the VARC-2 definition of AKI, AKR was defined as a decrease in serum creatinine ($\geq$50%) or $\geq$25% improvement in GFR up to 72 hours after TAVR.

### Results

AKI and AKR were respectively observed in 8.3 and 15.7% of the 574 patients included. AKI and AKR patients were associated to more advanced kidney disease at baseline. At a median follow-up of 608 days (range 355–893), AKI and AKR patients experienced an increased cardiovascular mortality compared to unchanged renal function patients (14.6% and 17.8% respectively, vs. 8.1%, CI 95%, p<0.022). Chronic kidney disease, (HR: 3.9; 95% CI 1.7–9.2; *p* < 0.001) was the strongest independent factor associated with AKI similarly to baseline creatinine level (HR: 1; 95% CI 1 to 1.1 *p* < 0.001) for AKR. 72-hours post procedural AKR (HR: 2.26; 95% CI 1.14 to 4.88; p = 0.021) was the strongest independent predictor of CV mortality.

**Data Availability Statement:** The data underlying the results presented in the study are the property of the Hopitaux Universitaires de Strasbourg. Furthermore, currently, we do not have

authorization to share any personal data with third external parties as the French legislation (Jardé law) does not allow a free sharing of human research participant data. Patients could be potentially identified based their age, sex, type of outcomes etc. in a single center study. Data could be made available upon request by submitting an email to President of the Scientific Board - Committee for Scientific Research of Hôpitaux Universitaires de Strasbourg, 1, place de l'hôpital, BP 426, 67091 Strasbourg cedex. Email: emmanuel.andres@chrustrasbourg.fr.

**Funding:** This work was supported by GERCA (Groupe pour l'Enseignement, la Recherche cardiologique en Alsace).

**Competing interests:** The authors have declared that no competing interests exist.

**Abbreviations:** ACE inhibitor, Angiotensin-converting enzyme inhibitor; AF, Atrial Fibrillation; AKI, Acute Kidney Injury; AKR, Acute Kidney Recovery; APT, Antiplatelet Therapy; ARBs, Angiotensin II Receptor Blockers; AS, Aortic stenosis; AVR, Aortic Valve Replacement; BMI, Body Mass Index; CAD, Coronary Artery Disease; CKD, Chronic Kidney Disease; COPD, Chronic Obstructive Pulmonary Disease; CRP, C-protein reactive; DAPT, Dual Antiplatelet Therapy; ECG, Electrocardiogram; EuroSCORE, Logistic EuroSCORE predicted risk of mortality at 30 days; GRF, Glomerular Filtration Rate; Hb, Haemoglobin Level; HF, Heart Failure; LV, Left Ventricle; LVEF, Left Ventricular Ejection Fraction; MACE, Major Adverse Cardiac Events; NOACs, Novel Oral Anticoagulant; TAVR, Transcatheter Aortic Valve Replacement; TTE, Transthoracic Echocardiography; VARC-2, Valve Academic Research Consortium-2 consensus.

## Conclusions

Both AKR and AKI negatively impact long term clinical outcomes of patients undergoing TAVR.

## Introduction

Acute kidney injury (AKI) is a frequent complication following transcatheter aortic valve replacement (TAVR) and is associated with poor prognosis. The reported incidence of post TAVR AKI is 22.1% ± 11.2 based on the Valve Academic Research Consortium-2 (VARC-2) definition [1,2]. Several risk factors of post procedural AKI have been identified including impaired baseline renal function, hemodynamic instability, sustained pacing, use of contrast medium, length of procedure and post procedural complications such as severe bleeding [3].

Acute kidney recovery (AKR), a phenomenon reverse to AKI has recently been investigated [4–6]. Indeed, TAVR represents a unique therapeutic modality in allowing instant reduction of the trans-aortic gradient, normalization of the aortic valve area and normalization of altered post-stenotic blood flow. In a kidney's perspective, rapid hemodynamic changes after TAVR including increased cardiac output, reduced left ventricular (LV) afterload and renal congestion may result in acute recovery of kidney function. So far, limited data exists on AKR and the entire spectrum of renal variations after TAVR including altered, unchanged or improved renal function. Therefore, this study aimed to investigate the incidence, predictors and prognostic impact of AKI and AKR amongst TAVR patients in a high-volume French hospital.

## Methods

### Patients

584 patients with severe aortic stenosis (AS) and high or intermediate surgical risk according to Logistic EuroSCORE were enrolled for TAVR at our institution (Nouvel Hôpital Civil, Strasbourg University, France) from November 2012 to May 2018. 10 patients with end-stage renal disease and ongoing dialysis were excluded. All participants gave their informed written consent and agreed to the anonymous processing of their data (France 2 registry IRB Information 911262). The FRANCE-2 registry (French Aortic National Corevalve and Edwards) is conducted by the French Society of Cardiology and the French Society of Thoracic and Cardiovascular Surgery. The study was approved by the CNIL's (Commission Nationale de l'Informatique et des Libertés) committee (ethical code number 911262).

### Definition of Acute kidney injury (AKI) and Acute kidney recovery (AKR)

Acute kidney injury (AKI) was defined according to the VARC-2 definition [1] as an absolute increase in serum creatinine ≥0.3 mg/dL (≥26.4 mmol/L) OR ≥50% increase in serum creatinine up to 72 hours after TAVR.

As there is currently no consensual definition of AKR, we partially followed the definition proposed by Azarbal et al. [4] and mirroring the VARC-2 definition of AKI. Therefore AKR was defined 1) as an absolute decrease in serum creatinine to ≥50% (≥0,50 decrease compared with baseline) up to 72 hours after the procedure OR 2) a ≥25% improvement in eGFR over 72 hours after the procedure or 3) a decrease of ≥0.3 mg/dL in serum creatinine over 72 hours after TAVR Patients with unchanged renal function were those who had neither AKI nor AKR post-TAVR.

### Collection of data and outcomes

All baseline and follow-up variables were recorded and entered into a secure, ethics-approved database. Creatinine was systematically collected up to 72 hours in all patients after TAVR. Clinical endpoint including mortality, stroke, bleeding, access-related complications and conduction disturbances were assessed according to the definitions provided by the VARC-2 guidelines. All clinical events were adjudicated by an events validation committee.

The primary endpoint of the study was the incidence of AKI and AKR 72 hours after the procedure. The secondary endpoints included all-cause mortality; a composite endpoint defined by cardiovascular mortality (defined as any death with demonstrable cardiovascular cause or any death that was not clearly attributable to a non-cardiovascular cause), stroke, myocardial infarction and rehospitalization for heart failure (defined as any event requiring the administration of intravenous therapy); and finally bleeding complications assessed according to VARC-2 definition and red blood cell transfusion ⎯ 2 Units requirement. These endpoints were compared across 3 categories: patients with AKI, AKR and unchanged renal function patients.

All patients were contacted by phone and questioned by a standardized questionnaire about their health status, symptoms, medications and the occurrence of adverse events.

### Statistical analysis

Quantitative variables were described according to AKI, AKR or unchanged renal function and expressed as means ± standard deviation. Categorical variables were expressed as counts and percentages. Categorical variables were compared with chi-square tests or Fisher's exact tests. Continuous variables were compared with the use of parametric (ANOVA) or non-parametric Mann-Whitney tests as appropriate. To determine predictors of AKI and AKR regression analysis was performed. Variables with $p < 0.05$ in univariate analysis were entered into a stepwise ascending multivariate analysis. Only one variable relating to chronic impairment of renal function was entered into the multivariate analysis model. Owing to collinearity between impairment of renal function and other parameters such as for example: EuroSCORE, chronic kidney disease etc. multivariate analysis was performed with only one variable relating to chronic renal function impairment. Calculations were performed using SPSS 17.0 for Windows (SPSS Inc., Chicago, IL, USA).

## Results

A total of 574 TAVR patients (mean age 83.1±7.4, 43.6% male, LVEF 54% and EuroScore II 22.8±14.4%) were included in the analysis. Mean baseline serum creatinine concentration was 112±52 μmol.L and 17.3% of the global cohort had chronic kidney disease (CKD) as defined by baseline creatinine>150umol/L. Most patients showed baseline CKD stage 3 (46% of the global cohort) and stage 4/5 (10.6% of the global cohort). Balloon and self-expandable devices were implanted in 352 (61.4%) and 222 (38.6%) patients respectively. No difference in contrast media volume administration was evidenced. Baseline, procedural and biological characteristics are summarized in Tables 1–3.

### Acute kidney injury, Acute kidney recovery and unchanged renal function

AKI was documented for 8.3%, AKR for 15.7% and unchanged renal function for 76% of the global cohort (Fig 1). Patients with AKI were associated with higher creatinine level at baseline (p<0.001) and more frequent chronic kidney disease as defined by serum creatinine level > 150umol/L (p<0.001) and history of atrial fibrillation (p = 0.07). No significant

**Table 1. Baseline characteristics.**

| | Global Cohort | Unchanged | AKI | AKR | p value |
|---|---|---|---|---|---|
| | n = 574 | n = 436 | n = 48 | n = 90 | |
| **Clinical parameters** | | | | | |
| Age—year | 83.1±7.4 | 83.1±7.4 | 84.5±4.8 | 82.4±8.4 | 0.28 |
| Male sex—no./total no. (%) | 250(43.6%) | 194(44.6%) | 24(50%) | 32(35.6%) | 0.19 |
| BMI–kg.m$^2$ | 26.8±6.5 | 26.6±5.3 | 25.8±4.9 | 28±9.6 | 0.73 |
| Hypertension | 432 (75.3) | 327 (75.7) | 37 (77.1) | 68 (75.6) | 0.948 |
| Diabetes mellitus | 154 (26.8) | 106 (24.3) | 16 (33.3) | 32 (35.6) | 0.051 |
| EuroSCORE (%) | 22.8±14.4 | 22±14.1 | 25.6±15.6 | 25.2±14.8 | 0.2 |
| COPD | 93 (16.3%) | 74(17.1%) | 10(20.8%) | 9(10%) | 0.17 |
| Stroke history | 81 (14.1%) | 58(13.3%) | 8(16.7%) | 15(16.7%) | 0.62 |
| AF history | 238(41.5%) | 170(39.1%) | 26(54.2%) | 42(46.7%) | 0.07 |
| NYHA II | 234(40.8%) | 185(42.5%) | 17(35.4%) | 32(35.6%) | 0.34 |
| NYHA III | 281(48.9%) | 211(48.3%) | 25(52.1%) | 45(50%) | 0.86 |
| NYHA IV | 59(10.3%) | 40(9.2%) | 6(12.5%) | 13(14.4%) | 0.29 |
| **Baseline ECG** | | | | | |
| Sinus rhythm | 421(73.5%) | 317(72.9%) | 37(77.1%) | 67(74.4%) | 0.8 |
| Paced rhythm | 40(7%) | 31(7.1%) | 4(8.3%) | 5(5.6%) | 0.81 |
| AF | 149(26%) | 115(26.4%) | 11(12.5%) | 23(25.6%) | 0.07 |
| LBBB | 101(17.6%) | 74(17%) | 10(20.8%) | 17(18.9%) | 0.76 |
| RBBB | 74(12.9%) | 52(12%) | 6(12.5%) | 16(17.8%) | 0.32 |
| **Baseline biological parameters** | | | | | |
| CKD (Creatinine>150umol/L) | 99(17.3%) | 52(12%) | 22(45.8%) | 25(27.8%) | <0.001 |
| Creatinine level (μmol.L) | 112±52 | 103±39 | 136±77 | 144±72 | <0.001 |
| eGRF Stage 1 -n (%) | 63(11) | 56 (12.9) | 6(12.5) | 1(1.1) | <0.001 |
| eGRF Stage 2 -n (%) | 187 (32.6) | 163 (37.5) | 9(18.8) | 15 (16.7) | |
| eGRF Stage 3A -n (%) | 149 (26) | 112 (25.7) | 11 (22.9) | 26 (28.9) | |
| eGRF Stage 3B -n (%) | 113 (19.7) | 79 (18.2) | 9 (18.8) | 25 (27.8) | |
| eGRF Stage 4 -n (%) | 55 (9.6) | 24 (5.5) | 12 (25) | 19 (21.1) | |
| eGRF Stage 5 -n (%) | 6 (1) | 1 (0.2) | 1 (2.1) | 4 (4.4) | |
| Hb (g/dL) | 12.2±1.7 | 12.1±1.7 | 12.6±1.6 | 12.1±1.5 | 0.24 |
| Platelets (10^9/L) | 224±73 | 225±71 | 218±86 | 225±74 | 0.82 |
| CT-ADP | 192±77 | 191±77 | 180±74 | 200±78 | 0.36 |
| **Echocardiography** | | | | | |
| LEVF (%) | 54±14 | 55±13 | 52±15 | 53±14 | 0.23 |
| LV mass—g.m$^2$ | 131±35 | 130±37 | 130±24 | 132±34 | 0.93 |
| LVendDV (mm) | 50±8 | 50±8 | 52±7 | 51±9 | 0.2 |
| Mean Gradient (mmHg) | 48±13 | 47±13 | 46±10 | 49±14 | 0.44 |
| Pulmonary arterial pressure | 40.7±14 | 40±13 | 41±13.5 | 42±16 | 0.62 |
| CT-derived aortic valve calcification score | 3388±1662 | 3348±1665 | 3392±1453 | 3580±1828 | 0.81 |

Abbreviations: AF: Atrial Fibrillation; AKI: Acute Kidney Injury; AKR: Acute Kidney Recovery; APT: Antiplatelets therapy; BMI: Body Mass Index; CKD: Chronic kidney disease; COPD: Chronic Obstructive Pulmonary Disease; CT: Computed tomography;. CT-ADP: Closure time of adenosine diphosphate; ECG: Electrocardiogram; eGRF: Estimated Glomerular Filtration Rate; Stage 1 with normal or high GFR (GFR > 90 mL/min); Stage 2 Mild CKD (GFR = 60–89 mL/min); Stage 3A Moderate CKD (GFR = 45–59 mL/min); Stage 3B Moderate CKD (GFR = 30-45mL/min); Stage 4 Severe CKD (GFR = 15–29 mL/min); Stage 5 End Stage CKD (GFR <15 mL/min); Hb: hemoglobin; LBBB: Left bundle branch block; LV: Left Ventricular; LVendDV: Left Ventricular diastolic diameter; LVEF: Left ventricular ejection fraction; NYHA: New York Heart Association functional class; RBBB: Right bundle branch block; TAVR: Transcatheter Aortic Valve Replacement.

**Table 2. Procedural characteristics.**

| | Global Cohort | Unchanged | AKI | AKR | p value |
|---|---|---|---|---|---|
| | n = 574 | n = 436 | n = 48 | n = 90 | |
| **Balloon valvuloplasty before TAVR** | 39(6.8%) | 27(6.2%) | 2(4.2%) | 10(11.1%) | 0.18 |
| **Approach** | | | | | |
| Transfemoral - no./total no. (%) | 512(89.5%) | 396(91%) | 38(80.9%) | 78(86.7%) | 0.061 |
| **Valve** | | | | | |
| Sapien - no./total no. (%) | 352(61.4%) | 276(63.4%) | 26(54.2%) | 50(55.6%) | 0.21 |
| Size Valve | | | | | |
| 23 mm - no./total no. (%) | 170(29.7%) | 131(30.1%) | 13(27.1%) | 26(28.9%) | 0.89 |
| 26 mm - no./total no. (%) | 196(34.2%) | 153(35.2%) | 15(31.3%) | 28(31.1%) | 0.68 |
| 29 mm - no./total no. (%) | 177(30.9%) | 132(30.3%) | 14(29.2%) | 31(34.4%) | 0.72 |
| 31 mm - no./total no. (%) | 21(3.7%) | 13(3%) | 5(10.4%) | 3(3.3%) | 0.034 |
| 34 mm - no./total no. (%) | 8(1.4%) | 5(1.1%) | 1(2.1%) | 2(2.2%) | 0.67 |
| Post Dilatation - no./total no. (%) | 61(10;6%) | 44(10.1%) | 5(10.4%) | 12(13.3%) | 0.67 |
| **Procedure** | | | | | |
| Contrast Volume (mL) | 159±56 | 157±54 | 177±62 | 159±56 | 0.11 |
| Procedure time (min) | 82±25 | 80±24 | 87±21 | 87±28 | 0.024 |

Abbreviations: AKI: Acute Kidney Injury; AKR: Acute Kidney Recovery; TAVR: Transcatheter Aortic Valve Replacement.

difference was noted according to the time of procedure (S1 Table). Post procedural bleeding, red blood cell transfusions and cardiovascular (CV) cause of death occurred more frequently in the AKI group. AKR patients experienced higher creatinine level at baseline. Similarly to AKI patients, the AKR group showed increased CV mortality rates compared to unchanged renal

**Table 3. Biological parameters.**

| | Global Cohort | Unchanged | AKI | AKR | p value |
|---|---|---|---|---|---|
| | n = 574 | n = 436 | n = 48 | n = 90 | |
| Creatinine level (μmol.L) | | | | | |
| Baseline | 112±52 | 103±39 | 136±77 | 144±72 | <0.001 |
| Post TAVR - Day 1 | 104±54 | 95±37 | 168±103 | 115±63 | <0.001 |
| Post TAVR - Day 3 | 106±54 | 98±39 | 177±89 | 106±63 | <0.001 |
| Hb (g/dL) | | | | | |
| Baseline | 12.2±1.7 | 12.1±1.7 | 12.6±1.6 | 12.1±1.5 | 0.24 |
| Post TAVR - Day 1 | 10.8±1.6 | 10.8±1.6 | 11.3±1.7 | 10.8±1.5 | 0.2 |
| Platelets ($10^9$/L) | | | | | |
| Baseline | 224±73 | 225±71 | 218±86 | 225±74 | 0.82 |
| Post TAVR - Day 1 | 178±60 | 178±57 | 180±70 | 177±64 | 0.96 |
| WBC Count ($10^9$/L) | | | | | |
| Baseline | 7.6±3.3 | 7.5±2.9 | 8.8±6.4 | 7.6±2.3 | 0.08 |
| Post TAVR - Day 1 | 9±3.4 | 9±3.5 | 8.8±3 | 9.2±3.3 | 0.77 |
| CRP (mg/L) | | | | | |
| Baseline | 9.8±12.2 | 9.2±11.1 | 12.4±13.4 | 11.2±15.9 | 0.17 |
| Post TAVR - Day 1 | 21.5±24.9 | 20.2±33.9 | 20.8±32.2 | 23.5±42.2 | 0.23 |

Abbreviations: AKI: Acute Kidney Injury; AKR: Acute Kidney Recovery; CRP: C-reactive protein; Hb: haemoglobin; TAVR: Transcatheter Aortic Valve Replacement; WBC: white blood cell.

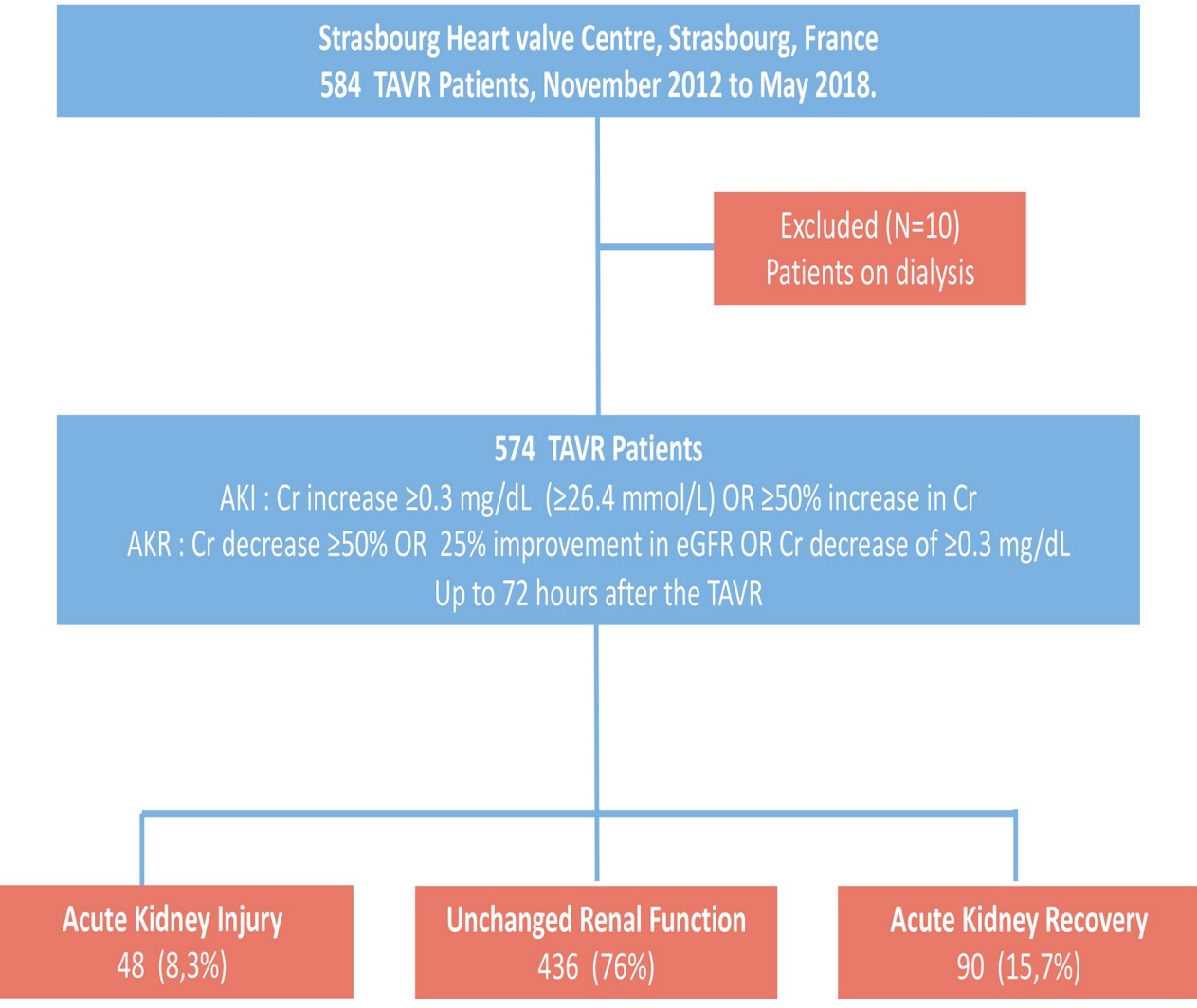

**Fig 1. Flow chart of the study.** Acute kidney injury and Acute kidney recovery following TAVR in a large cohort registry. Among 574 TAVR patients, Acute Kidney Injury (AKI) was documented for 8.3%, Acute Kidney Recovery (AKR) for 15,7% and unchanged renal function for 76% of the global cohort.

function patients. No significant differences in all-cause mortality nor other secondary MACE were evidenced between the three subsets of patients. All-cause mortality and cardiovascular events according according to renal function variations after TAVR are listed in Table 4.

At a median follow-up of 608 days (range 355–893), AKI and AKR patients experienced an increased cardiovascular mortality compared to unchanged renal function patients (14.6% and 17.8% respectively, vs. 8.1%, CI 95%, p<0.022) (Fig 2). No difference regarding all-cause mortality, non-CV death and MACE could be evidenced between the three subsets of patients (Fig 3A-3C).

## Predictors of AKI, AKR and cardiovascular mortality following TAVR

By univariate Cox analysis, EuroSCORE, chronic kidney disease (as defined by creatinine level>150μmol/L), baseline creatinine level, contrast volume, bleeding (both immediate and

**Table 4. Impact of Acute kidney injury and recovery on all-cause mortality and cardiovascular events.**

| | Global Cohort | Unchanged | AKI | AKR | p value |
|---|---|---|---|---|---|
| | n = 574 | n = 436 | n = 48 | n = 90 | |
| Death from any cause | 125(21.8%) | 86(19.8%) | 13(27.1%) | 26(28.9%) | 0.106 |
| CV Death | 58 (10.1%) | 44(8.1%) | 7(14.6%) | 16(17.8%) | 0.012 |
| Non-CV Death | 62(10.8%) | 50(11.5%) | 5(10.4%) | 7(7.8%) | 0.61 |
| Rehospitalization for HF | 103(18%) | 70(16.1%) | 11(22.9%) | 22(24.7%) | 0.101 |
| Myocardial Infarction | 14(2.4%) | 7(1.6%) | 3(6.3%) | 4(4.5%) | 0.056 |
| Stroke | 45(7.9%) | 29(6.7%) | 7(14.6%) | 9(10.1%) | 0.107 |
| MACE (Cardiovascular death, Rehospitalisation for HF, Stroke and/or infarct) | 186(32.4%) | 132(30.3%) | 17(35.4%) | 37(41%) | 0.125 |
| Post procedural bleeding | | | | | |
| Immediate all cause post procedural bleeding | 170(29.7%) | 113(26%) | 24(50%) | 33(36.7%) | 0.001 |
| Major and Life threating Bleeding | 68(11.9%) | 44(10.1%) | 11(22.9%) | 13(14.4%) | 0.024 |
| Life threating Bleeding | 31(5.4%) | 14(3.2%) | 7(14.6%) | 10(11.1%) | <0.001 |
| Red blood cell transfusion >2Units | 113(19.7%) | 68(15.6%) | 19(39.6%) | 26(28.9%) | <0.001 |
| Minor bleeding | 71(12.4%) | 55(12.6%) | 6(12.5%) | 10(11.1%) | 0.93 |
| Major and Life threating Bleeding during ICU stay | 99(17.3%) | 58(13.3%) | 18(37.5%) | 23(25.6%) | <0.001 |
| Bleeding at any time after discharge | 131(22.9%) | 89(20.5%) | 19(39.6%) | 23(25.6%) | 0.009 |

Abbreviations: AKI: Acute Kidney Injury; AKR: Acute Kidney Recovery; CV: Cardiovascular; HF: Heart failure; ICU: intensive care unit; MACE: major adverse cardiovascular events; TAVR: Transcatheter Aortic Valve Replacement.

major/life-threatening bleeding) were significant predictors of AKI (Table 5). In multivariate analysis, chronic kidney disease, (HR: 3.9; 95% CI 1.7–9.2; $p < 0.001$) remained the strongest independent factor associated with AKI.

By univariate Cox analysis, baseline creatinine level, procedure duration and major/ life-threatening bleeding were significant predictors of AKR (Table 6). In multivariate analysis, baseline creatinine level (HR: 1; 95% CI 1 to 1.1 $p < 0.001$) remained the sole independent predictor of AKR.

By univariate Cox analysis, chronic obstructive pulmonary disease (COPD), post procedural CRP level, and 72-hours post procedural AKR were significant predictors of CV mortality (Table 7). In multivariate analysis, COPD (HR: 2.4; 95% CI 1.17–4.95; $p = 0.017$) and 72-hours post procedural AKR (HR: 2.26; 95% CI 1.14 to 4.88; p = 0.021) remained strong independent of predictor of CV mortality.

## Discussion

To our knowledge, this is the first report highlighting a possible detrimental effect of post-TAVR acute kidney recovery. The salient results of the present study are as follows: (i) AKI was documented for 8.3% and AKR occurred in 15,7% of patients based upon Creatinine and/ or eGFR assessment and definitions, (ii) Both AKI and AKR had a dismal impact on CV mortality, (iii) baseline creatinine level was a strong and independent predictor associated to both AKI and AKR and finally (iv) 72-hours post procedural AKR was a strongest independent predictor of CV mortality.

### Acute kidney injury

Based on the VARC-2 definition [1], the reported incidence of post TAVR AKI is 22.1% ± 11.2 [2]. With 8,3% of AKI, our results are consistent with current existing evidence. Development

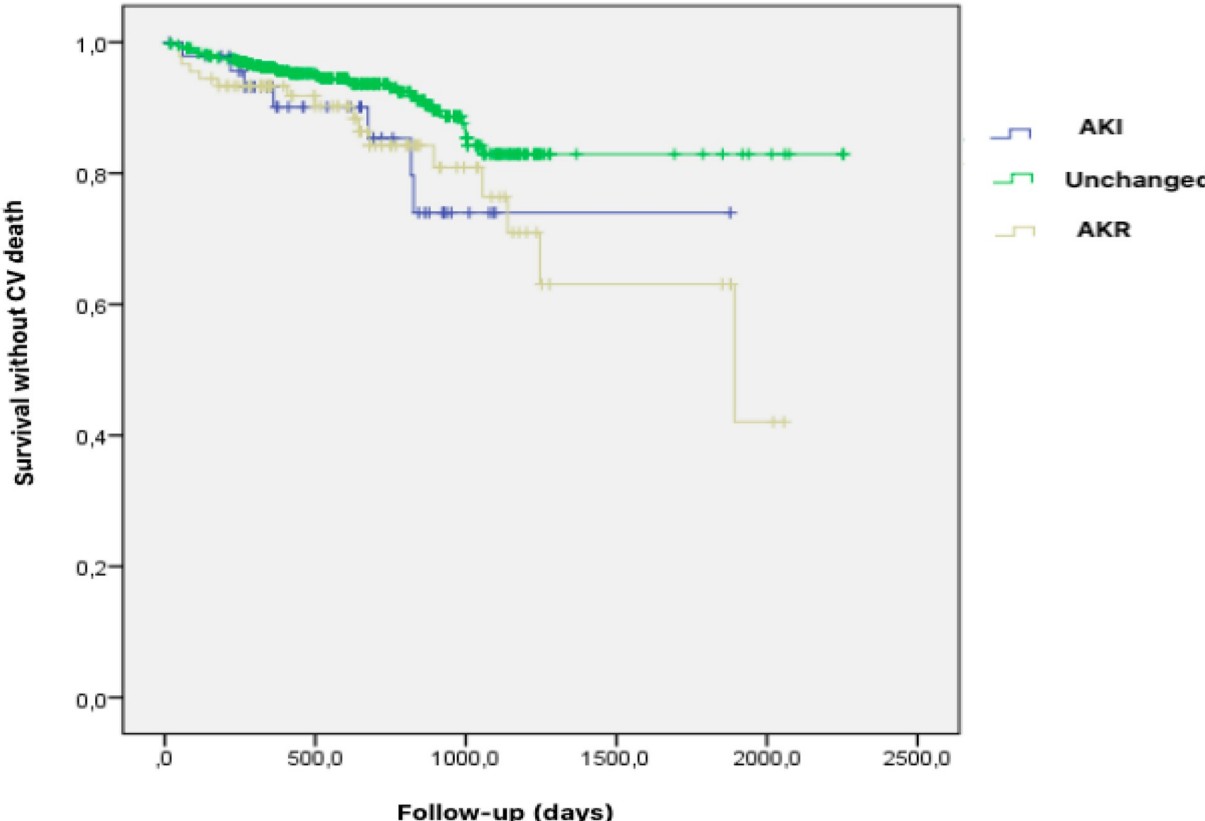

**Fig 2. Impact of Acute kidney injury (AKI), Acute kidney recovery (AKR) and unchanged renal function after TAVR on cardiovascular death.** Kaplan–Meier analysis for the probability of survival free from cardiovascular (CV) death after TAVR according to renal variations including altered (AKI), unchanged or improved (AKR) renal function. At a median follow-up of 608 days (range 355–893), AKI and AKR patients experienced an increased cardiovascular mortality compared to unchanged renal function patients (14,6% and 17,8% respectively, vs. 8,1%, CI 95%, p<0.022). **Abbreviations**: AKI = Acute Kidney Injury. AKR = Acute Kidney Recovery. CV = Cardiovascular. TAVR = Transcatheter Aortic Valve Replacement.

of AKI after TAVR was associated with altered baseline renal function and worse outcomes. Our findings regarding AKI are consistent with prior findings and reports suggesting a strong relationship between red blood cell transfusion, bleedings and vascular complications [7,8]. Conversely, our study did not support a strong relationship between contrast volume nor aortic severity (LVEF, mean aortic gradient) and the development of AKI.

## Acute kidney recovery: Conflicting results with regards to earlier studies

Despite very limited data existing on AKR after TAVR, our intriguing results with increased CV mortality linked to AKR challenge the current evidences supporting a beneficial impact of AKR after TAVR. Based on 1) a 25% improvement in eGFR over 48 hours after the procedure or 2) a decrease of ≥0.3 mg/dL in serum creatinine over 48 hours after TAVR, Azarbal et al [4] demonstrated that AKR occurred in sizeable proportion (32,5%) of TAVR patients with male sex, lack of chronic beta-blocker utilization and baseline CKD as sole independent predictors of AKR. Nijenhuis et al [5] referred to AKR as a post to pre-TAVR ratio of serum creatinine of ≤0.80. AKR was associated with a protective effect on two-year mortality (HR 0.53, 95%CI 0.30–0.93). Independent predictors of AKR were female gender, a preserved kidney function,

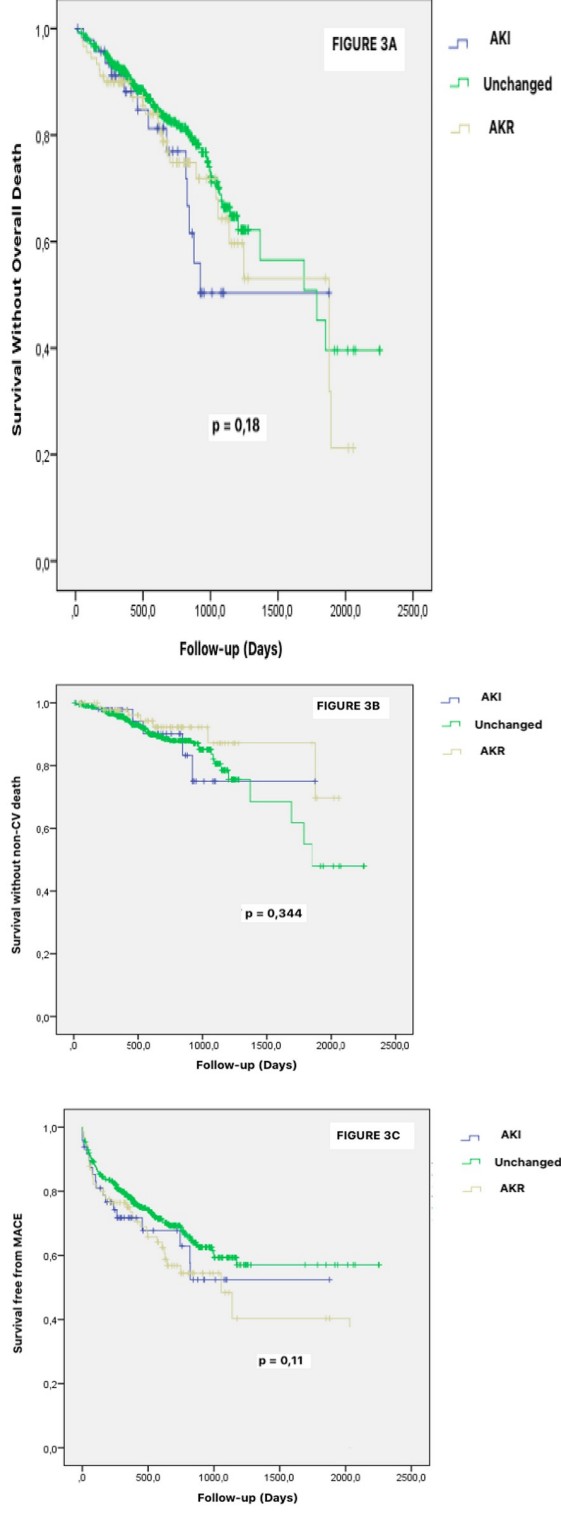

**Fig 3. Impact of Acute kidney injury (AKI), Acute kidney recovery (AKR) and unchanged renal function after TAVR on overall death, non-cardiovascular death and MACE.** Kaplan–Meier analysis for the probability of survival free overall death death (Fig 3A), non-cardiovascular death (Fig 3B) and MACE (Fig 3C) after TAVR according to renal variations including altered (AKI) and unchanged or improved (AKR) renal function. **Abbreviations**: AKI = Acute Kidney Injury. AKR = Acute Kidney Recovery. CV = Cardiovascular. TAVR = Transcatheter Aortic Valve Replacement.

**Table 5. Predictors of AKI after TAVR.**

| Predictors of AKI | | | |
|---|---|---|---|
| | **Univariate** | **p value** | **Multivariate** |
| | **HR (95% CI)** | | **HR (95% CI)** |
| **Baseline Characteristics** | | | |
| Age | 1.033 (0.987–1.082) | 0.17 | |
| Male sex | 1.317 (0.729–2.380) | 0.361 | |
| BMI | 0.97(0.83–1.1) | 0.69 | |
| EuroSCORE | 1.021 (1.004–1.039) | 0.015 | |
| Chronic Kidney disease KD (Cr >150μmol.L) | 4.934 (2.662–9.145) | <0.001 | 3.9(1.7–9.2) |
| Baseline Cr Level | 1(1–1.1) | 0.002 | |
| Baseline LEVF | 0.29 (0.3–2.4) | 0.24 | |
| Baseline Mean Aortic Gradient | 0.99 (0.97–1) | 0.45 | |
| **Procedural Characteristics** | | | |
| Balloon Predilatation prior to TAVR procedure (emergency procedure) | 0.57 (0.12–2.4) | 0.45 | |
| Transfemoral Approach | 0.453 (0.207–0.991) | 0.047 | 1(0.28–4.3) |
| Contrast volume | 1.1 (1–1.1) | 0.039 | 1.2(0.3–4.7) |
| Duration of procedure | 1.1 (0.9–1.1) | 0.13 | |
| **Post procedural events** | | | |
| Immediate bleeding | 2.6 (1.4–4.7) | 0.002 | |
| Major and life-threatening bleeding post TAVR | 3.3(1.7–6.2) | 0.016 | |

Abbreviations: AKI Acute Kidney Injury; AKR: Acute Kidney Recovery; CKD: Chronic Kidney Disease; Cr: Creatinine; LVEF: Left ventricular ejection fraction; TAVR: Transcatheter Aortic Valve Replacement.

Only one variable relating to chronic impairment of renal function was entered into the multivariate analysis model.

absence of atrial fibrillation and hemoglobin level. Both studies have stressed the importance of the pathophysiological mechanisms and the adaptive response to hemodynamic changes after TAVR that hint AKR. Indeed, in AS patients: volume overload, low systemic pressures combined with increased central venous and pulmonary artery pressures can compromise renal perfusion and result in type 2 cardiorenal syndrome [9,10].

As TAVR procedure represents a unique therapeutic modality in allowing instant reduction of the trans-aortic gradient, normalization of the aortic valve area and swift reduction of LV overload such interventional procedure offers a unique opportunity to monitor the cardio-renal interactions. We face that the rapid hemodynamic changes that offer TAVR such as correction of diastolic dysfunction, potential improved LVEF and/or cardiac output [11,12] are key determinants paving the way to the pathophysiological explanation of improved renal function in the AKR subgroup. Altogether, the present data identify AKR as a reversible cardiorenal syndrome with a detrimental effect of cardiorenal syndrome widely acknowledged [13,14]. Challenging the initial paradigm of a protective condition of AKR in TAVR, our insights have emphasized the potential noxious impact on CV death of renal reversal changes after TAVR.

## Acute kidney recovery: A kidney perspective of extra-aortic valve cardiac damage?

A new staging for the extent of extra-aortic valve cardiac damage in aortic stenosis was recently proposed by Généreux et al. [15]. This original staging consisted of four stage: Stage 0: no

**Table 6. Predictors of AKR after TAVR.**

**Predictors of AKI**

| | Univariate HR (95% CI) | p value | Multivariate HR (95% CI) | p value |
|---|---|---|---|---|
| **Baseline characteristics** | | | | |
| Age | 0.99 (0.96–1) | 0.424 | | |
| Male sex | 0.66 (0.42–1.1) | 0.085 | | |
| BMI | 1.03 (0.948–1.13) | 0.445 | | |
| EuroSCORE | 1 (0.99–1) | 0.24 | | |
| Chronic Kidney disease KD (Cr >150μmol.L) | 2.16 (1.29–3.63) | 0.003 | | |
| Baseline Cr Level | 1 (1–1.1) | 0.002 | 1 (1–1.1) | <0.001 |
| Baseline LEVF | 0.43 (0.089–2.1) | 0.3 | | |
| Baseline Mean Aortic Gradient | 1 (0.99–1) | 0.3 | | |
| **Procedural characteristics** | | | | |
| Balloon Predilatation prior to TAVR procedure (emergency procedure) | 1.87(0.88–3.99) | 0.1 | | |
| Transfemoral Approach | 0.75(0.38–1.48) | 0.41 | | |
| Contrast volume | 1 (0.99–1) | 0.96 | | |
| Duration of procedure | 1 (1–1.1) | 0.037 | 1 (0.99–1) | 0.37 |
| **Post procedural event** | | | | |
| Immediate bleeding | 1.54 (0.97–2.45) | 0.067 | | |
| Major life bleeding post TAVR | 2.63(1.19–5.79) | 0.016 | 2.2(0.83–6) | 0.109 |

Abbreviations: AKI: Acute Kidney Injury; AKR: Acute Kidney Recovery; CKD: Chronic Kidney Disease; Cr: Creatinine; LVEF: Left ventricular ejection fraction; TAVR: Transcatheter Aortic Valve Replacement.

Only one variable relating to chronic impairment of renal function was entered into the multivariate analysis model.

extravalvular cardiac damage; Stage 1: left ventricular damage; Stage 2: left atrial or mitral valve damage; Stage 3: pulmonary vasculature damage or significant tricuspid regurgitation and stage 4: right ventricular damage.

Each increment in stage was independently (HR 1.46, 95% CI 1.27–1.67, P < 0.0001) associated with increased mortality after aortic valve replacement (AVR). Our study surprisingly found that AKR was associated with increased CV mortality. To date, this novel approach of extra-valvular cardiac damage staging did not include hemodynamic parameters and cardiorenal interactions in particular. Given the interplay between worsening aortic stenosis and kidney function in cardiorenal syndrome, post TAVR AKR is likely to reflect a reversible cardiorenal syndrome and an "extra-cardiac damage" of AS. Importantly in the study by Généreux et al. [15], the extent of cardiac damage was the strongest predictor of adverse outcomes: normalization of the aortic valve area and hemodynamics instantly occur but the detrimental impact of extravalvular consequences of AS may persist despite AVR [16]. Aortic stenosis may contribute to cardiorenal syndrome that improves with TAVR, but our observation suggests that reversible post TAVR cardiorenal syndrome patients still have increased mortality. The role of extra-valvular cardiac damage staging in aortic valve stenosis management is becoming a topic of interest [17]. Future research may determine the benefit of early TAVR in asymptomatic patients with cardiorenal syndrome [18]. Further studies are also needed to test the incremental value of additional imaging parameters (e.g. intrarenal Doppler ultrasonography [19]), renal and systemic hemodynamic parameters as well as blood biomarkers (e.g. serum Cystatin C) to build an extra cardiac damage staging schemes in the field of TAVR.

**Table 7. Predictors of cardiovascular mortality.**

| Predictors of AKI | | | | |
|---|---|---|---|---|
| | Univariate | p value | Multivariate | p value |
| | HR (95% CI) | | HR (95% CI) | |
| **Baseline characteristics** | | | | |
| Age | 1.04 (0.99–1.09) | 0.066 | | |
| Male sex | 1.25 (0.74–2.09) | 0.39 | | |
| BMI | 0.92 (0.73–1.15) | 0.45 | | |
| COPD | 1.8 (1–3.25) | 0.049 | 2.4 (1.17–4.95) | 0.017 |
| EuroSCORE | 1 (0.98–1.02) | 0.81 | | |
| CKD (Cr level >150μmol.L) | 2.44 (1.41–4.19) | 0.001 | | |
| AF | 1.3 (0.74–2.31) | 0.35 | | |
| **Procedural characteristics** | | | | |
| Transfemoral Approach | 0.49 (0.25–0.96) | 0.04 | 0.53 (0.23–1.2) | 0.13 |
| **Post procedural characteristics** | | | | |
| Hemoglobin | 0.95 (0.79–1.14) | 0.6 | | |
| WBC count | 1.04 (0.97–1.13) | 0.27 | | |
| CRP | 1 (1.01–1.04) | 0.016 | 1.02 (1.01–1.04) | 0.063 |
| DAPT | 0.58 (0.35–0.98) | 0.04 | 0.55 (0.29–1.03) | 0.06 |
| Mean Aortic Gradient at one-month | 0.95 (0.89–1) | 0.12 | | |
| Aortic regurgitation >1/4 at one-month | 1.7 (0.94–3.28) | 0.07 | | |
| **Post procedural event** | | | | |
| Immediate bleeding | 0.81 (0.45–1.45) | 0.48 | | |
| AKI 72 hours | 1.79 (0.81–3.95) | 0.14 | | |
| AKR 72 hours | 1.82 (1.02–3.25) | 0.04 | 2.36 (1.14–4.88) | 0.021 |

Abbreviations: AF: Atrial Fibrillation; AKI: Acute Kidney Injury. AKR: Acute Kidney Recovery. CKD: Chronic Kidney Disease. COPD: Chronic Obstructive Pulmonary Disease. Cr: Creatinine; CRP: C-reactive protein; DAPT: Dual Antiplatelet Therapy; LVEF: Left ventricular ejection fraction; TAVR: Transcatheter Aortic Valve Replacement; WBC: White Blood Cell.

Only one variable relating to chronic impairment of renal function was entered into the multivariate analysis model.

## Study limitations

Several limitations should be taken into account in the interpretation of the data. Consensus on the definition of acute kidney recovery is still lacking and standardization of AKR definition would aid communication across the field. We did not challenge the association of acute kidney recovery with cardiac output improvement, and/or LV function recovery. Therefore, we cannot exclude the role of potential confounders, and in particular hemodynamic parameters, that were not accounted for in our model. The limited sample size in each e-GRF subgroup and the relative inhomogeneity in e-GRF subgroups add uncertainty to cardiovascular and renal endpoints, particularly because the pathophysiology of AKI/AKR may differ between groups. Finally, it is a single-centre study with all inherent limitations due to the design of such study.

## Conclusion

Both AKR and AKI negatively impact long term clinical outcomes of patients undergoing transcatheter aortic valve replacement. Both AKI and AKR patients experienced significantly higher cardiovascular mortality than those without a change in renal function. Pre-TAVR

CKD was the strongest independent predictor of post-TAVR AKI while seventy-two hours AKR was the strongest independent predictor of cardiovascular mortality".

## Supporting information

**S1 Table. AKI. AKR and unchanged renal function according to the time of procedure.** (DOCX)

## Author Contributions

**Conceptualization:** Olivier Morel.

**Data curation:** Marilou Peillex, Sebastien Hess, Antje Reydel, Marion Kibler, Adrien Carmona, Antonin Trimaille, Joe Heger, Hélène Petit-Eisenmann, Annie Trinh, Olivier Morel.

**Formal analysis:** Kensuke Matsushita, Laurence Jesel, Olivier Morel.

**Investigation:** Marilou Peillex, Benjamin Marchandot, Kensuke Matsushita, Eric Prinz, Sebastien Hess, Antje Reydel, Marion Kibler, Adrien Carmona, Antonin Trimaille, Joe Heger, Hélène Petit-Eisenmann, Annie Trinh, Patrick Ohlmann.

**Methodology:** Laurence Jesel, Olivier Morel.

**Project administration:** Olivier Morel.

**Resources:** Marilou Peillex, Eric Prinz, Sebastien Hess, Antje Reydel, Marion Kibler, Adrien Carmona, Antonin Trimaille, Joe Heger, Hélène Petit-Eisenmann, Annie Trinh, Olivier Morel.

**Supervision:** Olivier Morel.

**Validation:** Marilou Peillex, Benjamin Marchandot, Kensuke Matsushita, Eric Prinz, Laurence Jesel, Patrick Ohlmann, Olivier Morel.

**Writing – original draft:** Marilou Peillex, Benjamin Marchandot.

**Writing – review & editing:** Benjamin Marchandot, Kensuke Matsushita, Laurence Jesel, Patrick Ohlmann, Olivier Morel.

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
