## [Decision Letter · Decision Letter 0]

23 Feb 2021

PONE-D-21-00837

Acute kidney injury and Acute kidney recovery following Transcatether aortic valve replacement

PLOS ONE

Dear Dr. Marchandot,

Thank you for submitting your manuscript to PLOS ONE. After careful consideration, we feel that it has merit but does not fully meet PLOS ONE’s publication criteria as it currently stands. Therefore, we invite you to submit a revised version of the manuscript that addresses the points raised during the review process.

ACADEMIC EDITOR: All issues raised by expert reviewers are required.

We look forward to receiving your revised manuscript.

Kind regards,

Vincenzo Lionetti, M.D., PhD

Academic Editor

PLOS ONE

Journal Requirements:

2)  Thank you for including your ethics statement:  The approval for this study was obtained from the France 2 study 911262

3) In ethics statement in the manuscript and in the online submission form, please provide additional information about the patient records/samples used in your retrospective study. Specifically, please ensure that you have discussed whether all data/samples were fully anonymized before you accessed them. We have noted that the current study is a retrospective cohort study, however we have noted that written informed consent was obtained from participants. This suggests that the patients were prospectively identified. Please provide some further clarification on this. 

4) Please include additional information regarding the survey or questionnaire used in the study and ensure that you have provided sufficient details that others could replicate the analyses. For instance, if you developed a questionnaire as part of this study and it is not under a copyright more restrictive than CC-BY, please include a copy, in both the original language and English, as Supporting Information.

Furthermore, in your Methods section, please provide a justification for the sample size used in your study, including any relevant power calculations (if applicable).

5)  Thank you for stating the following in the Acknowledgments/Funding Section of your manuscript:

[This work was supported by GERCA (Groupe pour l’Enseignement, la

Recherche Cardiologique en Alsace).]

 [The author(s) received no specific funding for this work.]

6)  We note that you have indicated that data from this study are available upon request. PLOS only allows data to be available upon request if there are legal or ethical restrictions on sharing data publicly. For information on unacceptable data access restrictions, please see http://journals.plos.org/plosone/s/data-availability#loc-unacceptable-data-access-restrictions.

7) Please amend either the title on the online submission form (via Edit Submission) or the title in the manuscript so that they are identical.

8) Please include captions for your Supporting Information files at the end of your manuscript, and update any in-text citations to match accordingly. Please see our Supporting Information guidelines for more information: http://journals.plos.org/plosone/s/supporting-information.

9) Thank you for submitting the above manuscript to PLOS ONE. During our internal evaluation of the manuscript, we found significant text overlap between your submission and the following previously published works, some of which you are an author.

https://www.mdpi.com/2077-0383/9/4/905/htm (Introduction, paragraph 1; Methods)

https://www.dovepress.com/acute-kidney-injury-after-transcatheter-aortic-valve-replacement-in-th-peer-reviewed-fulltext-article-CIA (Acute Kidney Recovery: Conflicting results with regards to earlier studies, paragraph 1, sentence 2)

https://www.sciencedirect.com/science/article/pii/S0735109719353987?via%3Dihub (Acute Kidney Recovery: a kidney perspective of extra-aortic valve cardiac damage?, paragraph 1)

Please revise the manuscript to rephrase the duplicated text, cite your sources, and provide details as to how the current manuscript advances on previous work. Please note that further consideration is dependent on the submission of a manuscript that addresses these concerns about the overlap in text with published work.

Reviewers' comments:

Reviewer's Responses to Questions

**Comments to the Author**

1. Is the manuscript technically sound, and do the data support the conclusions?

Reviewer #1: Partly

Reviewer #2: Partly

2. Has the statistical analysis been performed appropriately and rigorously? 

Reviewer #1: Yes

Reviewer #2: Yes

3. Have the authors made all data underlying the findings in their manuscript fully available?

Reviewer #1: Yes

Reviewer #2: Yes

4. Is the manuscript presented in an intelligible fashion and written in standard English?

Reviewer #1: Yes

Reviewer #2: Yes

5. Review Comments to the Author

Reviewer #1: This single center retrospective study on acute kidney injury and acute kidney recovery following transcatether aortic valve replacement is a very interesting study which has several points of strength but also many limitations. The number of patients studied is high, data reported have been almost well presented and the discussion section focuses on interesting hypothesis on the cardiorenal relationship and the global outcomes after correcting aortic stenosis.

Major criticisms

1. Patients of this study are elderly (mean age 83.1±7.4 years) and most of them at baseline show CKD stage 3 (about 46% of the global cohort) and stage 4/5 (about 10% of the global cohort). Age and CKD stage are an independent risk factor for post-surgical AKI and CV mortality. This should be highlighted and not only resumed in a supplementary table. Moreover, the prevalence of diabetes mellitus and hypertension is not described in the baseline characterization of the population studied, and also those factors are independent factors for post-surgical AKI and CV mortality. It would be very interesting to analyze the post TAVR data on serum creatinine and CV risk in those subgroups of patients.

2. Although the definition of acute kidney recovery is not widely accepted and has not been standardized, it could represent an interesting focus in this kind of procedure, in which global hemodynamics change immediately after the surgical procedure, involving heart, vessels and kidneys. However, the main limitation of this study is that there are no data on renal haemodynamics before and after TAVR, thus weakening the results and therefore the discussion section. The renal microcirculation status before and after TAVR has not been studied (for example with colorDoppler evaluation of renal resistance indexes). In renal artery stenosis, if renal microcirculation is compromised independently of CKD stage, the surgical correction with angioplasty does not improve renal perfusion. On the contrary, angioplasty may determine in those cases a reperfusion damage, thus worsening kidney function and increasing CV risk. Moreover, serum creatinine levels vary depending also on systemic blood pressure and hydration status, and no data on these factors before and after TAVR have been reported. This could influence the evaluation of AKI and/or AKR after TAVR.

Patients of this study are more likely to have a compromised renal microcirculation due to age, CKD and systemic damage related to aortic stenosis, so it is pivotal to evaluate it for a correct interpretation of the data reported.

Minor criticisms:

1. In the brief summary the acronym TAVR has been cited without the extended form.

2. In the “definition of AKR” section it is stated that patients with unchanged renal function were those who had neither AKI nor AKR post-TAVR. 12% of them have CKD (as reported in table 1). It would be interesting to analyze this subset of patients and their outcome based on CKD stage.

3. The last sentence before the “study limitation” section is not clear (Following Généreux’ staging precept, AKR is likely to reflect an “extra-cardiac damage”. Still reversible. Hence, some will argue that fixed cardiorenal syndrome patients after TAVR should have a worse outcome compared to reversal counterparts. As fixed cardiorenal syndrome patients are part of “unchanged renal function”, their worse prognosis is likely to be soften. Moreover, the underlying process is complex and the cardiorenal syndrome is difficult to define and identify because it encompasses complex multifactorial facets) is not very clear.

4. In table 1 (biological parameters), what CT-ADP stands for?

Reviewer #2: The authors wrote an interesting research on the incidence of AKI and ARK in a population of patients undergoing TAVR procedure.

The results are very interesting and deserving of disclosure but the manuscript presents multiple problems so it should be revised.

First of all, the multiplicity of subparagraphs is a distraction for the reader, it would be better to combine some of them in order to make the reading less jagged.

Starting from the abstract the conclusions are telegraphic, please review.

Highlight: Add a highlight on Aki which should become the first point.

In the introduction we pass directly from bibliography 2 to 4, the number 3 has been lost, therefore useful revision of the numbering of the bibliography.

Methods:

Compared to the purposes stated in the introduction, different primary endpoints are then declared in the collection of data paragraph, the latter therefore should all become secondary endpoints.

Results: still too many sub-paragraphs, the basal characteristics are well reported in tables 1 and 2 but poorly commented in the results paragraph where instead the main result of the study is immediately reported.

In fact, this result is not a baseline characteristic of the patient and should be reported later in Impact of AKI ...., also moving the table 3

In the paragraph Impact of Aki ... the subdivision of patients into 3 groups appears, a subdivision not mentioned before, it should therefore be explained in the methods as well as in the results.

Discussion: Also in this paragraph the number of subparagraphs should be reduced.

In the sub-paragraph AKI we talk about complications such as the number of transfusions without having mentioned them in methods or results but they are shown only in the table

Tables 1, 3,4 show p values but it is not easy to understand which groups they refer to, as the table lacks asterisks or symbols that serve as a reference.

The figure chosen as central represents the patient selection flowchart that would go among the results as fig. 1

Again the conclution are telegraphic.

6. PLOS authors have the option to publish the peer review history of their article (what does this mean?). If published, this will include your full peer review and any attached files.

Reviewer #1: No

Reviewer #2: No

---

## [Author Response · Author response to Decision Letter 0]

20 Jun 2021

The third wave of the COVID-19 pandemic has brought tough challenges to Strasbourg’s Cardiology Team. As these difficult times required more time for us to complete our response, we could not return our paper in a timely manner. We would like to apologize to both the PLOS one editorial board and Reviewers.

Sincerely yours,

Olivier Morel MD, PhD

Point-by-point response to the editorial board 

We thank the editorial board for the constructive suggestions. All line numbers and pages mentioned in the upcoming responses refer to the tracked changes made effective in the revisited manuscript.

On behalf of all authors, 

Olivier Morel

Journal Requirements: 

Change has been made according to the journal production team’s request

2) Thank you for including your ethics statement: The approval for this study was obtained from the France 2 study 911262 

The French Aortic National CoreValve and Edwards (FRANCE 2) Registry was established under the French Societies of Cardiology and of Thoracic and Cardiovascular Surgery. All patients provided written informed consent before undergoing the procedure, including consent for anonymous processing of their data. The registry was approved by the institutional review board of the French Ministry of Health. Full Protocol available : Gilard M, Eltchaninoff H, Iung B, et al. Registry of transcatheter aortic-valve implantation in high-risk patients. N Engl J Med. 2012

It is now clearly stated in the method section

All participants gave their informed written consent and agreed to the anonymous processing of their data (France 2 registry IRB Information 911262). 

3) In ethics statement in the manuscript and in the online submission form, please provide additional information about the patient records/samples used in your retrospective study. Specifically, please ensure that you have discussed whether all data/samples were fully anonymized before you accessed them. We have noted that the current study is a retrospective cohort study, however we have noted that written informed consent was obtained from participants. This suggests that the patients were prospectively identified. Please provide some further clarification on this. 

All TAVR patients included in this study are part of the global TAVR cohort performed in France, as listed in the FRANCE 2 registry, and were prospectively included in the study. Patients were enrolled at 34 centers including ours, Strasbourg University Hospital

It is now clearly stated in the method section

All participants gave their informed written consent and agreed to the anonymous processing of their data (France 2 registry IRB Information 911262). 

The typo error regarding the retrospective nature of the study has been modified. 

4) Please include additional information regarding the survey or questionnaire used in the study and ensure that you have provided sufficient details that others could replicate the analyses. For instance, if you developed a questionnaire as part of this study and it is not under a copyright more restrictive than CC-BY, please include a copy, in both the original language and English, as Supporting Information. 

Furthermore, in your Methods section, please provide a justification for the sample size used in your study, including any relevant power calculations (if applicable). 

-As disclosed in the manuscript; follow-up variables were recorded and entered into a database. All patients were contacted by phone and questioned by a standardized questionnaire about their health status, symptoms, medications, the occurrence of adverse events, and the treatment about adverse events according to the FRANCE 2 protocol (Full Protocol available : Gilard M, Eltchaninoff H, Iung B, et al. Registry of transcatheter aortic-valve implantation in high-risk patients. N Engl J Med. 2012)

-No sample size/power calculation is applicable to this observational cohort study

5) Thank you for stating the following in the Acknowledgments/Funding Section of your manuscript: 

[This work was supported by GERCA (Groupe pour l’Enseignement, la Recherche Cardiologique en Alsace).] 

[The author(s) received no specific funding for this work.]

Change has been made according to the journal production team’s request

6) We note that you have indicated that data from this study are available upon request. PLOS only allows data to be available upon request if there are legal or ethical restrictions on sharing data publicly. For information on unacceptable data access restrictions, please see http://journals.plos.org/plosone/s/data-availability#loc-unacceptable-data-access-restrictions. 

b) If there are no restrictions, please upload the minimal anonymized data set necessary to replicate your study findings as either Supporting Information files or to a stable, public repository and provide us with the relevant URLs, DOIs, or accession numbers. Please see http://www.bmj.com/content/340/bmj.c181.long for guidelines on how to de-identify and prepare clinical data for publication. For a list of acceptable repositories, please see http://journals.plos.org/plosone/s/data-availability#loc-recommended- repositories. 

To date, the dataset includes potentially identifying patient informations; de-identified data have been updated so far. Therefore we have modified our previous statement

7) Please amend either the title on the online submission form (via Edit Submission) or the title in the manuscript so that they are identical. 

Change has been made according to the journal production team’s request

8) Please include captions for your Supporting Information files at the end of your manuscript, and update any in-text citations to match accordingly. Please see our Supporting Information guidelines for more information: http://journals.plos.org/plosone/s/supporting-information

Change has been made according to the journal production team’s request

9) Thank you for submitting the above manuscript to PLOS ONE. During our internal evaluation of the manuscript, we found significant text overlap between your submission and the following previously published works, some of which you are an author. 

https://www.mdpi.com/2077-0383/9/4/905/htm (Introduction, paragraph 1; Methods) https://www.dovepress.com/acute-kidney-injury-after-transcatheter-aortic-valve-replacement-in-th-peer-reviewed-fulltext-article- CIA (Acute Kidney Recovery: Conflicting results with regards to earlier studies, paragraph 1, sentence 2) https://www.sciencedirect.com/science/article/pii/S0735109719353987?via%3Dihub (Acute Kidney Recovery: a kidney perspective of extra-aortic valve cardiac damage?, paragraph 1) 

We would like to make you aware that copying extracts from previous publications, especially outside the methods section, word- for-word is unacceptable. In addition, the reproduction of text from published reports has implications for the copyright that may apply to the publications. 

Please revise the manuscript to rephrase the duplicated text, cite your sources, and provide details as to how the current manuscript advances on previous work. Please note that further consideration is dependent on the submission of a manuscript that addresses these concerns about the overlap in text with published work. 

We thank the editorial board for bringing this point to our attention. As noted, text overlap between this submission and previous works are related to previous papers published by our research team, including 

(i) A former paper published by our team in JCM (Peillex M et al. Bedside Renal Doppler Ultrasonography and Acute Kidney Injury after TAVR. J Clin Med. 2020). The first paragraph of the introduction has been edited to dismiss any conflicts between the two papers, despite same authorship.

(ii) A poster presentation submitted to the French Society of Cardiology Congress earlier this year (M. Peillex et al. Acute kidney injury and Acute kidney recovery following TAVR: Conflicting results with regards to earlier studies, Archives of Cardiovascular Diseases Supplements, Volume 13, Issue 1, 2021)

(iii) The discussion has been re-edited according to both reviewer 1&2 concerns and accurate copyright updated

Point-by-point response to the comments of Reviewer #1

We thank the reviewer for her/his constructive suggestions and improved manuscript as a result. All line numbers and pages mentioned in the upcoming responses refer to the tracked changes made effective in the revisited manuscript.

On behalf of all authors, 

Olivier Morel

Reviewer #1: This single center retrospective study on acute kidney injury and acute kidney recovery following transcatheter aortic valve replacement is a very interesting study which has several points of strength but also many limitations. The number of patients studied is high, data reported have been almost well presented and the discussion section focuses on interesting hypothesis on the cardiorenal relationship and the global outcomes after correcting aortic stenosis.

Major criticisms

1. Patients of this study are elderly (mean age 83.1±7.4 years) and most of them at baseline show CKD stage 3 (about 46% of the global cohort) and stage 4/5 (about 10% of the global cohort). Age and CKD stage are an independent risk factor for post- surgical AKI and CV mortality. This should be highlighted and not only resumed in a supplementary table. Moreover, the prevalence of diabetes mellitus and hypertension is not described in the baseline characterization of the population studied, and also those factors are independent factors for post-surgical AKI and CV mortality. It would be very interesting to analyze the post TAVR data on serum creatinine and CV risk in those subgroups of patients. 

We thank the reviewer because these points are useful to clarify missing practical information regarding baseline characteristics of our cohort. First, diabetes and hypertension status at baseline are missing in the proposed manuscript. Therefore, we followed the reviewer’s request and disclose these two missing variables in table 1. These clinical characteristics do not appear to significantly differ between groups and thus were not included in a multivariate analysis. Moreover, age, diabetes on insulin etc.. are already part of the EUROSCORE composite score; and in order to avoid overfitting in the model, hypertension, hypertension (p=0.948) and diabetes (p=0.051) were not chosen for multivariable analysis

Second, AKI and AKR according to baseline renal function are now reported in table 1. As noted by the reviewer, most patients showed baseline CKD stage 3 (46% of the global cohort) and stage 4/5 (10.6% of the global cohort). Indeed, (i) the observational nature of our study, (ii) the “limited” sample size in each e-GRF subgroup and (iii) the relative inhomogeneity in e-GRF subgroups add uncertainty to CV and renal endpoints, particularly because the pathophysiology of AKI/AKR may differ between groups. To acknowledge these facts, we added a statement to the limitation section.

Changes

To accommodate both reviewer 1&2, an extensive editing of the results and limitations section was performed

(i) Results section: 

A total of 574 TAVR patients (mean age 83.1±7.4, 43.6% male, LVEF 54% and EuroScore II 22.8±14.4%) were included in the analysis. Mean baseline serum creatinine concentration was 112±52 µmol.L and 17.3% of the global cohort had chronic kidney disease (CKD) as defined by baseline creatinine>150umol/L. Most patients showed baseline CKD stage 3 (46% of the global cohort) and stage 4/5 (10.6% of the global cohort). Balloon and self-expandable devices were implanted in 352 (61.4%) and 222 (38.6%) patients respectively. No difference in contrast media volume administration was evidenced. Baseline, procedural and biological characteristics are summarized in Table 1, 2 and 3.

(ii) Table 1

Hypertension, Diabetes mellitus, AKI and AKR according to baseline renal function are now reported in table 1. 

Of note no statistically significant difference between the groups with respect to AKI/AKR/unchanged renal function was observed for the hypertensive status, though there was a trend towards a statistical significance regarding diabetes status. 

Table 1. Baseline characteristics 

 Global Cohort Unchanged AKI AKR p value

 n=574 n=436 n=48 n=90 

Clinical parameters 

Hypertension 432 (75.3) 327 (75.7) 37 (77.1) 68 (75.6) 0.948

Diabetes mellitus 154 (26.8) 106 (24.3) 16 (33.3) 32 (35.6) 0.051

Table 1. Baseline characteristics 

 Global Cohort Unchanged AKI AKR p value

 n=574 n=436 n=48 n=90 

Baseline biological parameters 

CKD (Creatinine>150umol/L) 99(17.3%) 52(12%) 22(45.8%) 25(27.8%) <0.001

Creatinine level (µmol.L) 112±52 103±39 136±77 144±72 <0.001

eGRF Stage 1 -n (%) 63(11) 56 (12.9) 6(12.5) 1(1.1) 

<0.001 

eGRF Stage 2 -n (%) 187 (32.6) 163 (37.5) 9(18.8) 15 (16.7) 

eGRF Stage 3A -n (%) 149 (26) 112 (25.7) 11 (22.9) 26 (28.9) 

eGRF Stage 3B -n (%) 113 (19.7) 79 (18.2) 9 (18.8) 25 (27.8) 

eGRF Stage 4 -n (%) 55 (9.6) 24 (5.5) 12 (25) 19 (21.1) 

eGRF Stage 5 -n (%) 6 (1) 1 (0.2) 1 (2.1) 4 (4.4) 

(iii) Study limitations section: 

“The limited sample size in each e-GRF subgroup and the relative inhomogeneity in e-GRF subgroups add uncertainty to cardiovascular and renal endpoints, particularly because the pathophysiology of AKI/AKR may differ between groups”

2. Although the definition of acute kidney recovery is not widely accepted and has not been standardized, it could represent an interesting focus in this kind of procedure, in which global hemodynamics change immediately after the surgical procedure, involving heart, vessels and kidneys. However, the main limitation of this study is that there are no data on renal haemodynamics before and after TAVR, thus weakening the results and therefore the discussion section. The renal microcirculation status before and after TAVR has not been studied (for example with colorDoppler evaluation of renal resistance indexes). In renal artery stenosis, if renal microcirculation is compromised independently of CKD stage, the surgical correction with angioplasty does not improve renal perfusion. On the contrary, angioplasty may determine in those cases a reperfusion damage, thus worsening kidney function and increasing CV risk. Moreover, serum creatinine levels vary depending also on systemic blood pressure and hydration status, and no data on these factors before and after TAVR have been reported. This could influence the evaluation of AKI and/or AKR after TAVR.

Patients of this study are more likely to have a compromised renal microcirculation due to age, CKD and systemic damage related to aortic stenosis, so it is pivotal to evaluate it for a correct interpretation of the data reported.

We fully agree with the questions and limitations raised by the reviewer.

First, we fully agree that the lack of standardization with regard to the definition of AKR is a major issue and presents confusion and difficulty in inter-cohort comparisons, hindering widespread adoption of AKR a dedicated post TAVR endpoint in upcoming trials and studies. 

Second, we recently aimed to elucidate the association of renal resistance index (RRI) and cardio-renal hemodynamics with acute kidney injury (AKI) after TAVR (Peillex et al. J Clin Med). This pilot study showed that higher post-procedural RRI represents an important and independent predictive factor of AKI. TAVR patients exhibited higher baseline RRI values (0,76±0,7) compared to normal known and accepted values: 0.60 ± 0.10 in adults due to arterial stiffness and RRI was the result of a complex interaction between intrarenal circulation and systemic hemodynamics. Albeit limited to a small sample size, we provided data suggesting that higher RRI is linked to transient higher valvuloarterial impedance (Zva) and total arterial load (TAL) one day after TAVR. 

Changes: To take into account the Reviewer’s comment, the discussion has been revised and the pivotal role of post TAVR cardiorenal hemodynamics mentioned in the manuscript as part of future applications. 

Changes

Discussion section

Acute Kidney Recovery: a kidney perspective of extra-aortic valve cardiac damage?

Recently, Généreux et al. (15) proposed a new staging for the extent of extra-aortic valve cardiac damage in AS: no extravalvular cardiac damage (Stage 0), left ventricular damage (Stage 1), left atrial or mitral valve damage (Stage 2), pulmonary vasculature damage or significant tricuspid regurgitation (Stage 3), or right ventricular damage (Stage 4). The extent of cardiac damage was independently (HR 1.46 per each increment in stage, 95% CI 1.27–1.67, P < 0.0001) associated with increased mortality after aortic valve replacement (AVR). Our study surprisingly found that AKR was associated with increased CV mortality. To date, this novel approach of extra-valvular cardiac damage staging did not include hemodynamic parameters and cardiorenal interactions in particular. Given the interplay between worsening aortic stenosis and kidney function in cardiorenal syndrome, post TAVR AKR is likely to reflect a reversible cardiorenal syndrome and an “extra-cardiac damage” of AS. Importantly in the study by Généreux et al. (15), the extent of cardiac damage was the strongest predictor of adverse outcomes: normalization of the aortic valve area and hemodynamics instantly occur but the detrimental impact of extravalvular consequences of AS may persist despite AVR. 

Aortic stenosis may contribute to cardiorenal syndrome that improves with TAVR, but our observation suggests that reversible post TAVR cardiorenal syndrome patients still have increased mortality. The role of extra-valvular cardiac damage staging in aortic valve stenosis management is becoming a topic of interest (16). Future research may determine the benefit of early TAVR in asymptomatic patients with cardiorenal syndrome (17). Further studies are also needed to test the incremental value of additional imaging parameters (e.g. intrarenal Doppler ultrasonography (18)), renal and systemic hemodynamic parameters as well as blood biomarkers (e.g. serum Cystatin C) to build an extra cardiac damage staging schemes in the field of TAVR

Minor criticisms:

1. In the brief summary the acronym TAVR has been cited without the extended form.

Change has been made according to the reviewer’s request

2. In the “definition of AKR” section it is stated that patients with unchanged renal function were those who had neither AKI nor AKR post-TAVR. 12% of them have CKD (as reported in table 1). It would be interesting to analyze this subset of patients and their outcome based on CKD stage.

We fully agree with this comment and thank the reviewer for bringing this critical point to our attention. Investigating incidence, predictors, and outcomes of AKR following TAVR is interesting from a clinical perspective because we often observe impressive improvements in chronically impaired renal function after this procedure, but this phenomenon is inconsistent and unpredictable across patients. More likely as shown in our study and acknowledged by the reviewer in CKD stage 3 patients. However, a limitation of focusing only on “unchanged renal function” patients in the present additional analysis proposed by the reviewer is that this group may exhibit both acute improvements in renal function and a "masked" concomitant transient contrast-induced AKI… Indeed, we first sought to make the present statistical analysis proposed by the reviewer while writing the present paper, but we felt that any conclusions made out such analysis may be too tricky to interpreted do to several issues :

Small sample size : 12% with CKD out of the subgroup unchanged renal function group

Unchanged group : not change at all OR improved renal function masked by a concomitant transient contrast-induced AKI..

Therefore, we faced that conducting such analysis may be very confusing and “too much”hypothesis-generating.

3. The last sentence before the “study limitation” section is not clear (Following Généreux’ staging precept, AKR is likely to reflect an “extra-cardiac damage”. Still reversible. Hence, some will argue that fixed cardiorenal syndrome patients after TAVR should have a worse outcome compared to reversal counterparts. As fixed cardiorenal syndrome patients are part of “unchanged renal function”, their worse prognosis is likely to be soften. Moreover, the underlying process is complex and the cardiorenal syndrome is difficult to define and identify because it encompasses complex multifactorial facets) is not very clear.

We would like to thank the Reviewer for her/his comment. To accommodate the Reviewer comment, the discussion has been revised (as follow): 

Changes

Discussion section

Acute Kidney Recovery: a kidney perspective of extra-aortic valve cardiac damage?

Recently, Généreux et al. (15) proposed a new staging for the extent of extra-aortic valve cardiac damage in AS: no extravalvular cardiac damage (Stage 0), left ventricular damage (Stage 1), left atrial or mitral valve damage (Stage 2), pulmonary vasculature damage or significant tricuspid regurgitation (Stage 3), or right ventricular damage (Stage 4). The extent of cardiac damage was independently (HR 1.46 per each increment in stage, 95% CI 1.27–1.67, P < 0.0001) associated with increased mortality after aortic valve replacement (AVR). Our study surprisingly found that AKR was associated with increased CV mortality. To date, this novel approach of extra-valvular cardiac damage staging did not include hemodynamic parameters and cardiorenal interactions in particular. Given the interplay between worsening aortic stenosis and kidney function in cardiorenal syndrome, post TAVR AKR is likely to reflect a reversible cardiorenal syndrome and an “extra-cardiac damage” of AS. Importantly in the study by Généreux et al. (15), the extent of cardiac damage was the strongest predictor of adverse outcomes: normalization of the aortic valve area and hemodynamics instantly occur but the detrimental impact of extravalvular consequences of AS may persist despite AVR. 

Aortic stenosis may contribute to cardiorenal syndrome that improves with TAVR, but our observation suggests that reversible post TAVR cardiorenal syndrome patients still have increased mortality. The role of extra-valvular cardiac damage staging in aortic valve stenosis management is becoming a topic of interest (16). Future research may determine the benefit of early TAVR in asymptomatic patients with cardiorenal syndrome (17). Further studies are also needed to test the incremental value of additional imaging parameters (e.g. intrarenal Doppler ultrasonography (18)), renal and systemic hemodynamic parameters as well as blood biomarkers (e.g. serum Cystatin C) to build an extra cardiac damage staging schemes in the field of TAVR

4. In table 1 (biological parameters), what CT-ADP stands for? 

We thank the reviewer for bringing this point to our attention. CT-ADP indicates closure time of adenosine diphosphate; and it is now clearly stated in the abbreviation section of table 1. Our group has recently demonstrated that CT-ADP > 180 sec, a surrogate marker of HMW-VWF defect, as measured during the time course of the procedure allows an accurate identification of PVL in patients undergoing TAVR and is a major predictor of early and late bleeding (Kibler et al. JACC; Matsushita et al. Thromb Haemost) No difference regarding CT-ADP was observed between the 3 subsets of patients (AKI, AKR, unchanged renal function) in the present study.

Point-by-point response to the comments of Reviewer #2

We thank the reviewer for her/his constructive suggestions and improved manuscript as a result. All line numbers and pages mentioned in the upcoming responses refer to the tracked changes made effective in the revisited manuscript.

On behalf of all authors, 

Olivier Morel

Reviewer #2: The authors wrote an interesting research on the incidence of AKI and ARK in a population of patients undergoing TAVR procedure. The results are very interesting and deserving of disclosure but the manuscript presents multiple problems so it should be revised. 

First of all, the multiplicity of subparagraphs is a distraction for the reader, it would be better to combine some of them in order to make the reading less jagged.

We thank the reviewer for bringing this critical point to our attention. To accommodate the reviewer, an extensive editing of the manuscript was performed

Starting from the abstract the conclusions are telegraphic, please review.

Change has been made according to reviewer’s request

Changes

Abstract section

“Both AKR and AKI negatively impact long term clinical outcomes of patients undergoing TAVR”

Conclusion section

“Both AKR and AKI negatively impact long term clinical outcomes of patients undergoing transcatheter aortic valve replacement. Both AKI and AKR patients experienced significantly higher cardiovascular mortality than those without a change in renal function. Pre-TAVR CKD was the strongest independent predictor of post-TAVR AKI while seventy-two hours AKR was the strongest independent predictor of cardiovascular mortality.”

Highlight: Add a highlight on Aki which should become the first point. 

Change has been made according to the reviewer’s request

Changes: Highlights section

- AKI occurred in 8.3% of 574 TAVR patients 

- AKR occurred in 15.7% 

- AKI and AKR patients experienced increased cardiovascular mortality 

- Baseline creatinine level was the strongest predictor of AKR.

- The prognostic value of acute kidney recovery (AKR) remains under debate

In the introduction we pass directly from bibliography 2 to 4, the number 3 has been lost, therefore useful revision of the numbering of the bibliography.

We thank the reviewer for bringing this critical point to our attention. The typo error has been corrected.

Changes: Introduction section

Line 10. Depicting the scope of AKI in the field of TAVR relies on a variety of factors from impaired baseline renal function, hemodynamic instability during pacing, use of contrast medium to post procedural complications such as bleeding (3).

Methods:

Compared to the purposes stated in the introduction, different primary endpoints are then declared in the collection of data paragraph, the latter therefore should all become secondary endpoints. 

We thank the reviewer for bringing this critical point to our attention. To accommodate the Reviewer comment, the methods has been revised (as follow): 

Changes: Methods section - Collection of Data and Outcomes

Collection of Data and Outcomes

All baseline and follow-up variables were recorded and entered into a secure, ethics-approved database. Creatinine was systematically collected up to 72 hours in all patients after TAVR. Clinical endpoint including mortality, stroke, bleeding, access-related complications and conduction disturbances were assessed according to the definitions provided by the VARC-2 guidelines. All clinical events were adjudicated by an events validation committee. 

The primary endpoint of the study was the overall all-cause mortality after TAVR. The secondary endpoints were a composite endpoint defined by cardiovascular mortality (defined as any death with demonstrable cardiovascular cause or any death that was not clearly attributable to a non-cardiovascular cause), stroke and rehospitalization for heart failure (defined as any event requiring the administration of intravenous therapy). 

The primary endpoint of the study was the incidence of AKI and AKR 72 hours after the procedure. The secondary endpoints included all-cause mortality; a composite endpoint defined by cardiovascular mortality (defined as any death with demonstrable cardiovascular cause or any death that was not clearly attributable to a non-cardiovascular cause), stroke, myocardial infarction and rehospitalization for heart failure (defined as any event requiring the administration of intravenous therapy); and finally bleeding complications assessed according to VARC-2 definition and red blood cell transfusion ⩾ 2 Units requirement. These endpoints were compared across 3 categories: patients with AKI, AKR and unchanged renal function patients.

All patients were contacted by phone and questioned by a standardized questionnaire about their health status, symptoms, medications and the occurrence of adverse events.

Changes: Tables section. The tittle of table 4 has been rephrased 

Table 4. Impact of Acute Kidney injury and Recovery on All-cause Mortality and Cardiovascular Events

Results: still too many sub-paragraphs, the basal characteristics are well reported in tables 1 and 2 but poorly commented in the results paragraph where instead the main result of the study is immediately reported.

In fact, this result is not a baseline characteristic of the patient and should be reported later in Impact of AKI ...., also moving the table 3 

To accommodate the reviewer

- The headline “baseline Characteristics” has been removed 

- Several sub-paragraphs headlines have been removed 

- To accommodate both reviewer 1&2, an extensive editing of the results section was performed

Changes

(i) Results section: 

A total of 574 TAVR patients (mean age 83.1±7.4, 43.6% male, LVEF 54% and EuroScore II 22.8±14.4%) were included in the analysis. Mean baseline serum creatinine concentration was 112±52 µmol.L and 17.3% of the global cohort had chronic kidney disease (CKD) as defined by baseline creatinine>150umol/L. Most patients showed baseline CKD stage 3 (46% of the global cohort) and stage 4/5 (10.6% of the global cohort). Balloon and self-expandable devices were implanted in 352 (61.4%) and 222 (38.6%) patients respectively. No difference in contrast media volume administration was evidenced. Baseline, procedural and biological characteristics are summarized in Table 1, 2 and 3.

In the paragraph Impact of Aki ... the subdivision of patients into 3 groups appears, a subdivision not mentioned before, it should therefore be explained in the methods as well as in the results.

Changes: To take into account the Reviewer’s comment, the methods section has been extensively edited and one specific sentence regarding the subdivision in 3 groups added to the manuscript 

Changes

(i) Methods section: 

These endpoints were compared across 3 categories: patients with AKI, AKR and unchanged renal function patients.

Definitions of AKI, AKR and unchanged renal function patients are given in the methods section and results for each group disclosed in the results section

Discussion: Also, in this paragraph the number of subparagraphs should be reduced.

In the sub-paragraph AKI we talk about complications such as the number of transfusions without having mentioned them in methods or results, but they are shown only in the table 

Following the reviewers’ request, details of bleedings endpoints definition and RBC requirements are now clearly stated in the methods sections and one sentence has been added to the results section.

(i) Methods section: 

“and finally bleeding complications assessed according to VARC-2 definition and red blood cell transfusion ⩾ 2 Units requirement”

(i) Results section: 

Post procedural bleeding, red blood cell transfusions and cardiovascular (CV) cause of death occurred more frequently in the AKI group

The figure chosen as central represents the patient selection flowchart that would go among the results as fig. 1

Change has been made according to the reviewer’s request

Again the conclution are telegraphic. 

Change has been made according to reviewer’s request

Changes

Abstract section

“Both AKR and AKI negatively impact long term clinical outcomes of patients undergoing TAVR”

Conclusion section

“Both AKR and AKI negatively impact long term clinical outcomes of patients undergoing transcatheter aortic valve replacement. Both AKI and AKR patients experienced significantly higher cardiovascular mortality than those without a change in renal function. Pre-TAVR CKD was the strongest independent predictor of post-TAVR AKI while seventy-two hours AKR was the strongest independent predictor of cardiovascular mortality.”

---

## [Decision Letter · Decision Letter 1]

26 Jul 2021

Acute kidney injury and Acute kidney recovery following Transcatether aortic valve replacement

PONE-D-21-00837R1

Dear Dr. Marchandot,

We’re pleased to inform you that your manuscript has been judged scientifically suitable for publication and will be formally accepted for publication once it meets all outstanding technical requirements.

Kind regards,

Vincenzo Lionetti, M.D., PhD

Academic Editor

PLOS ONE

Additional Editor Comments (optional):

Reviewers' comments:

Reviewer's Responses to Questions

**Comments to the Author**

1. If the authors have adequately addressed your comments raised in a previous round of review and you feel that this manuscript is now acceptable for publication, you may indicate that here to bypass the “Comments to the Author” section, enter your conflict of interest statement in the “Confidential to Editor” section, and submit your "Accept" recommendation.

Reviewer #1: All comments have been addressed

2. Is the manuscript technically sound, and do the data support the conclusions?

Reviewer #1: Yes

3. Has the statistical analysis been performed appropriately and rigorously? 

Reviewer #1: Yes

4. Have the authors made all data underlying the findings in their manuscript fully available?

Reviewer #1: Yes

5. Is the manuscript presented in an intelligible fashion and written in standard English?

Reviewer #1: Yes

6. Review Comments to the Author

Reviewer #1: The authors have answered to all the questions in a satisfactory way and have properly changed the manuscript, so in my opinion it is now suitable for publication

7. PLOS authors have the option to publish the peer review history of their article (what does this mean?). If published, this will include your full peer review and any attached files.

Reviewer #1: No

---

## [Editor Report · Acceptance letter]

29 Jul 2021

PONE-D-21-00837R1 

Acute Kidney Injury and Acute Kidney Recovery Following Transcatheter Aortic Valve Replacement 

Dear Dr. Marchandot:

I'm pleased to inform you that your manuscript has been deemed suitable for publication in PLOS ONE. Congratulations! Your manuscript is now with our production department. 

Kind regards, 

on behalf of

Prof. Vincenzo Lionetti 

Academic Editor

PLOS ONE